# A human cerebral organoid model of West Nile virus encephalitis shows innate immunocompetency

Johanna Friederike Steffen [1] ✉, Lina Widerspick [1,2], Stephanie Jansen [1,3,4] & Dennis Tappe[1,4]

West Nile virus (WNV), an arbovirus of emerging global interest, can cause neuroinvasive disease in humans. Currently, no protective vaccine or specific treatment is available for human WNV encephalitis. The virus induces neuronal cell death, while astrocytes and microglia cells are suspected to contribute to WNV pathology. Hence, understanding their role is crucial for future treatment approaches. In this study, we establish a WNV encephalitis model using human cerebral organoids, generated with male iPSCs. Infection results in heterogeneous kinetics with an early strong replication potentially leading to viral clearance, while a late peak was associated with more long-term infection. Viral foci are seen in cortical-like areas, rich in neurons and astrocytes, however void of microglia. Pro-inflammatory cytokines (IL-6, TNF-α, IL-18), chemokines (CXCL10, CCL17, CX3CL1, CCL2) and biomarkers (IL-1RA, sTREM-1, sRAGE, BDNF) are increasingly released. Conclusively, human cerebral organoids make suitable WNV encephalitis models with valuable properties to study acute and long-term infection.

Encephalitis, the inflammation of brain tissue, is a serious condition that affects the health of adults and children causing morbidity, often high mortality, as well as permanent neurological disability[1]. The condition manifests with various symptoms such as headache, fever, a plethora of neurological dysfunctions including coma, and is often resulting from the infection with various neurotropic viruses[2,3]. Despite its severe implications, specific treatment options are rare, demonstrating a substantial gap of knowledge which ultimately leaves most patients to only supportive care[2]. West Nile virus (WNV), an arbovirus of the family *Flaviviridae*, is one of the causes of life-threatening encephalitis in humans[4]. Still, in humans, WNV neuroinvasive disease, as most neurotropic viral infections[2], can neither be prevented by a vaccine nor treated with any specific antiviral treatment[5]. Besides a mortality of 10 to 15%, WNV encephalitis frequently results in long-term sequelae, highlighting its implications for human health[2,4,6].

Since the first discovery of WNV in Uganda in 1937[7], it is nowadays of global interest, as it has spread to Europe, the Middle East, North America, and West Asia[5]. WNV has become endemic throughout the United States of America[4], with more than 30,000 cases of WNV neuroinvasive disease until 2024[8,9] showing the rapid geographical spread of this virus since its first detection on the continent in an outbreak in New York city in 1999[10]. Similarly, following the first European outbreak of WNV in France in 1962[11], human autochthonous WNV infection is reported in many European countries today. Furthermore, besides its main vector, *Culex* mosquitos, WNV has been detected in a broad range of mosquitoes[12,13], highlighting the potential for further geographical expansion of WNV and consequently an increase in human WNV encephalitis cases.

The knowledge on neuropathology caused by WNV and the inflammatory response in the human brain is very limited. Nevertheless, autopsy studies revealed neuronal damage, the formation of

[1]Bernhard Nocht Institute for Tropical Medicine, Hamburg, Germany. [2]German Center for Infection Research (DZIF), Partner site Hamburg-Lübeck-Borstel-Riems, Hamburg, Germany. [3]Faculty of Mathematics, Informatics and Natural Sciences, University of Hamburg, Hamburg, Germany. [4]These authors contributed equally: Stephanie Jansen, Dennis Tappe. ✉e-mail: johanna.steffen@bnitm.de

microglial nodules and astrocytosis alongside with invading immune cells[14–17], which is generally in line with a strong inflammation of the central nervous system (CNS). Moreover, besides neurons, astrocytes have been observed as a WNV target cell population[17]. The analysis of cerebrospinal fluid (CSF) of WNV patients revealed the presence of several pro-inflammatory cytokines and chemokines, such as IL-6, CCL2, and CXCL10[18,19]. Detailed insights into the cellular determinants of WNV encephalitis remain limited to animal models, or in vitro models of human or rodent cell monocultures. Briefly, neuronal cell death was reported and the release of various pro-inflammatory cytokines was proven to be important for fighting WNV infection[20]. Nevertheless, despite the favorable outcome of a local pro-inflammatory response, it is generally known in the context of encephalitis that such responses can become overwhelming and thus contribute to pathology[21]. Indeed, astrocytes and microglia, as local immunocompetent cells of the CNS with the potential to induce or amplify such pro-inflammatory processes, have been suspected to contribute not only to the antiviral defense but also WNV pathology and long-term neurological deficits[6,22]. Hence, studying the role of astrocytes and microglia is crucial to understand underlying mechanisms of WNV encephalitis, thereby contributing to future therapy approaches.

This study reports the establishment of a WNV encephalitis model using human cerebral organoids containing neuronal, microglia and astrocyte populations to address the disease in a complex in vitro system.

## Results

### Human cerebral organoids are susceptible to WNV and show diverse courses of infection

In this study, mature 100-days-old human cerebral organoids were infected with WNV with a low infectious dose of $1 \times 10^4$ viral particles per organoid. We found WNV to robustly infect human cerebral organoids, releasing high amounts of infectious particles, whereby individual infection kinetics of infected organoids showed considerable heterogeneity (Fig. 1a). To further investigate the course of infection, organoids that remained within the experiment for a minimum of 7 dpi were analyzed for their peak in viral titer. Strikingly, the viral peak days displayed a bimodal distribution (Fig. 1b). Many organoids reached the viral peak within the first days of the experiment, leading to an overall median peak day of 4 dpi. For most of the remaining organoids that did not peak until day 4 post infection, we observed a viral peak towards later stages of the experiment, until 14 dpi. Furthermore, the viral titers reached at peak showed a high variance (Fig. 1c). While the median is observed at $2.18 \times 10^6$ FFU/ml, the viral peak titers ranged from $1.57 \times 10^2$ FFU/ml to $8.85 \times 10^8$ FFU/ml. Despite these heterogeneous courses if infection, it can be noted that neither the WNV-infected organoids nor the uninfected control showed signs of an excessive cytopathic effect (CPE), such as disintegration and visible debris in the supernatant, during the experiment.

Interestingly, in some of the organoids we observed a drop in viral titer below the limit of detection (LOD) of 88.5 FFU/ml, with no subsequent increase until the end of the experiment, indicating viral

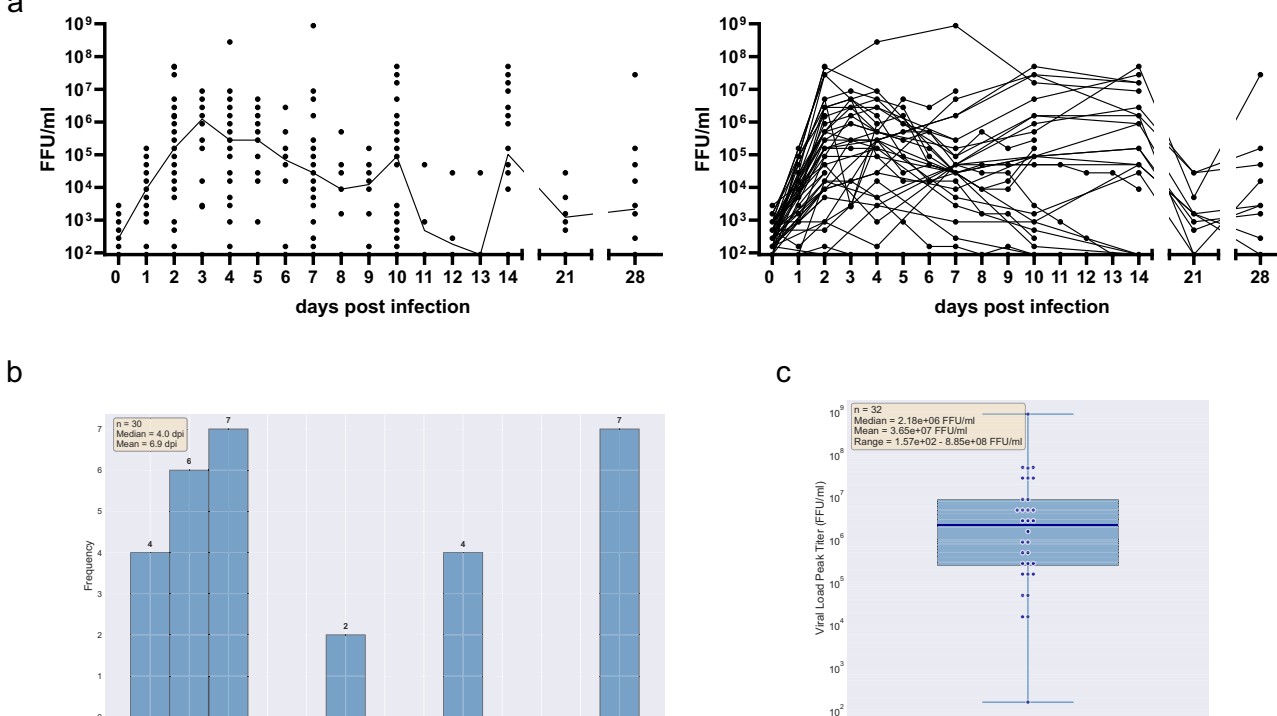

**Fig. 1 | WNV infection of human cerebral organoids.** Human cerebral organoids were infected with $1 \times 10^4$ WNV infectious particles. Results were obtained in two independent experiments, both sampling to at least 14 dpi, whereby one experiment additionally included 21 and 28 dpi. WNV infection of human cerebral organoids resulted in robust infection. **a** Heterogeneous courses of infection were observed over the time of 28 days. $n = 4$–47 per timepoint (see Methods for details; Exact sample sizes are provided in Source Data file). Left: Line trough median. Right: Individual infection kinetics. **b** Individual courses of infection varied in regards to the day the peak viral titer was reached, uncovering a bimodal distribution. Peak day was evaluated for organoids with minimum 7 dpi in the experiment. Two organoids did not show one clear peak, but reached peak titer over several time points and were therefore excluded from this analysis. ($n = 30$). **c** Peak titer was evaluated for all organoids with minimum 7 dpi in the experiment ($n = 32$). Titers reached at the viral peak showed a median value of $2.18 \times 10^6$ FFU/ml with a broad range from $1.57 \times 10^2$ FFU/ml to $8.85 \times 10^8$ FFU/ml. Individual data points as dots. Box shows median and interquartile range; whiskers show minimum and maximum values. Limit of detection (LOD): $88.5 \times 10^2$ FFU/ml. Source data are provided as a Source Data file.

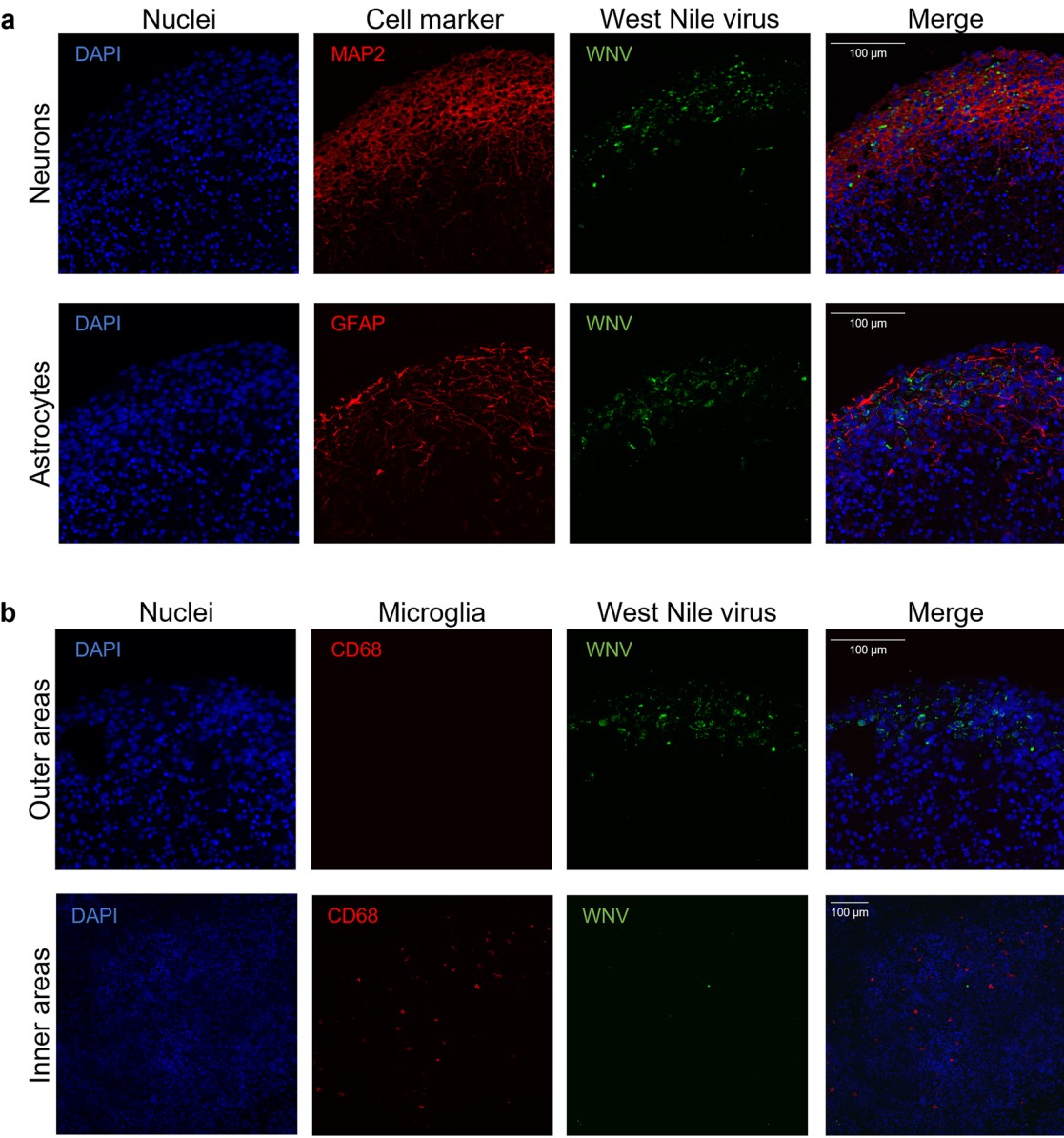

**Fig. 2 | WNV infection localizes to outer cortical-like areas of human cerebral organoids.** Immunofluorescence staining of WNV-infected human cerebral organoids for WNV envelope protein, adult neurons (MAP2), astrocytes (GFAP) or microglia (CD68) and cell nuclei (DAPI). Depicted is one infected organoid at 14 dpi. Similar staining patterns were observed in two additional organoids. **a** The staining revealed WNV envelope protein in small foci in outer areas of the organoids. These areas were observed to be rich in adult neurons and astrocytes. **b** No microglia were detected in the areas positive for WNV envelope protein. Instead, singular microglia were observed throughout the organoids. Scale bars, 100 μm. Source data are provided as a Source Data file.

clearance. This was observed at various time points from 8 to 28 dpi, with a median clearance day of 14 dpi. Considering all organoids that were tested for at least 14 days, a clearance rate of about 37.5% (9/24) was observed.

## WNV infection localizes to outer cortical-like areas of human cerebral organoids

In order to uncover potential targets of WNV infection in human cerebral organoids, infected and uninfected organoids were harvested at 2, 4, 7, 10, and 14 dpi, as well as later time points at 21 and 28 dpi. Notably, the WNV-infected organoids released varying amounts of infectious particles at the time of harvesting. Hence, some organoids were harvested while no infectious particles could be detected in the supernatant anymore, while others had viral titers up to $1.57 \times 10^7$ FFU/ml at harvest.

Organoids positive for WNV envelope protein by immunofluorescence staining showed viral titers above $2.8 \times 10^5$ FFU/ml in the supernatant. In contrast, no viral envelope protein was detected in organoids which had no infectious particles in the supernatant at the time of sampling. However, a high titer in the supernatant did not ensure positive WNV staining in the organoid sections tested. In organoids positive for WNV envelope protein, immunostaining revealed relatively small areas of the outer cortical-like layer of the organoids to be affected (Fig. 2a).

WNV-positive loci were rich in neuronal cells (MAP2+) and astrocytes (GFAP+), potentially revealing these cell types to be the WNV targets in this model. Furthermore, the viral protein was predominantly found in the perinuclear regions. As the cell markers used in this study are found in the cell body of the respective cell type, a full overlap in markers is not to be expected. Microglia could not be

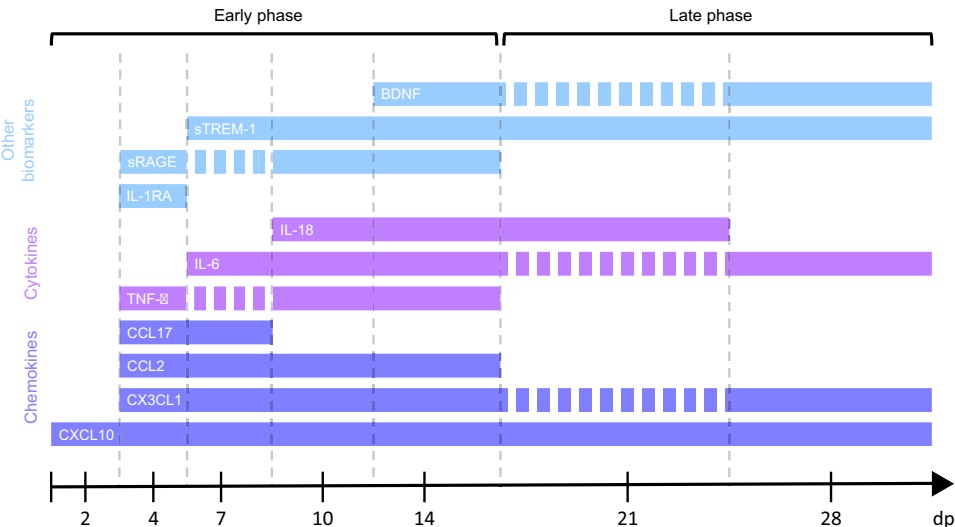

**Fig. 3 | Temporal release of diverse biomarkers by WNV-infected human cerebral organoids.** Screening of supernatants of WNV-infected human cerebral organoids for cytokines, chemokines and other biomarkers in the early phase until 14 dpi (2 independent experiments) and the late phase until 28 dpi (1 experiment) using a bead-based LegendPlex™ assay. Early phase and late phase responses were analyzed individually using a mixed-effects model (REML) with Šídák's multiple comparisons method for the differences between infected organoids and uninfected controls. A statistically significant increase is marked by a solid line ($p < 0.05$). Non-significant results between significant comparisons are marked by a dashed line ($p > 0.05$). C-C motif chemokine 17 (CCL17). C-C motif chemokine 2 (CCL2). C-X3-C motif chemokine 1 (CX3CL1). C-X-C motif chemokine 10 (CXCL10). Brain-derived neurotropic factor (BDNF). Interleukin 1 receptor antagonist protein (IL-1RA). Interleukin 6 (IL-6). Interleukin 18 (IL-18). Soluble receptor for advanced glycosylation end products (sRAGE). Soluble triggering receptor expressed on myeloid cells 1 (sTREM-1). Tumor necrosis factor alpha (TNF-α). Controls $n = 11–45$ per timepoint. Infected $n = 12–47$ per timepoint (see Methods for details). Source data are provided as a Source Data file. Exact sample sizes are provided in Source Data file.

detected in areas positive for WNV, but single microglia were shown to be present throughout the more inner areas of the organoids without positive staining of viral envelope protein (Fig. 2b). The identity of the labelled cell types was confirmed by additional markers, NeuN (neurons), S100B (astroctyes), CX3CR1 (microglia) (Supplementary Fig. 1). Importantly, microglia were observed to be heterogeneously distributed throughout uninfected organoids, including inner less organized areas and outer cortical-like layers (Supplementary Fig. 1a), as well as infected organoids (Supplementary Fig. 1b).

Thus, there was neither evidence of microglia infection nor attraction/migration towards the foci of WNV infection in the cerebral organoid model.

## The neuroinflammatory profile of human cerebral organoids captures diverse aspects of WNV encephalitis

The response of human cerebral organoids to infection with WNV was characterized by the release of secreted cytokines, chemokines, and other biomarkers at several time points during the early phase of infection (2 to 14 dpi) and at two later time points in the late phase (21 and 28 dpi). First, the comparison of WNV-infected organoids to uninfected controls revealed the chemokine CXCL10 to show the earliest onset for an increased release of all markers included upon 2 dpi (Fig. 3). This effect was observed throughout all time points of the early and late phase. Next, the chemokines CCL17, CCL2 and CX3CL1 were observed to be increasingly released by WNV-infected organoids starting at 4 dpi. Thereby, the increase of CX3CL1 could be observed not only in the early, but also in the late phase, as seen also for CXCL10. In contrast, CCL17 and CCL2 only increased during the early phase, but not later on. Notably, CCL2 increased until the last time point of the early phase (14 dpi), while the increased release of CCL17 could only be observed until 7 dpi. Moreover, the pro-inflammatory cytokines IL-6, TNF-α and IL-18 were increasingly released upon WNV infection. Here, TNF-α showed the earliest onset starting at 4 dpi, lasting throughout the early phase, with no increased release in the late phase. While the onset of an increased IL-6 and IL-18 release was at a later time point,

starting at 7 and 10 dpi respectively, this effect was observed beyond the early phase. Furthermore, the biomarkers IL-1RA, sRAGE, sTREM-1 and BDNF were observed to be increasingly released by WNV-infected organoids. Interestingly, IL-1RA was overall the only marker observed to be increasingly released at only one of the included time points, namely 4 dpi. While the onset of an increased sRAGE release was observed at the same time, this effect lasted until the end of the early phase. On the other hand, the increase of sTREM-1, starting at 7 dpi, was observed throughout the early but also the late phase. The overall latest onset was observed for BDNF at 14 dpi and lasted beyond the early phase. For detailed information on the biomarker release in the early and late phase, see Supplementary Figs. 2–5.

## Organoid morphology and course of infection associate with differences in the neuroinflammatory profile

As previously described, we observed heterogeneous courses of WNV infection in human cerebral organoids, which displayed a bimodal distribution of the day at which viral titers reached their peak. In order to investigate whether the course on infection associates with patterns in the neuroinflammatory profile, we applied a two-component Gaussian mixture model (GMM) to partition the dataset based on the course of infection, defined by the day of the viral peak. Two organoids were excluded from this analysis as they did not reach one clear peak but reached their highest viral titer at several peaks (For details see Methods). Those organoids were manually assigned to Type B as their course of infection is more closely represented by these organoids. Subsequently, organoids with an early peak in viral titer until 4 dpi are further referred to as Type A, while organoids with a late peak in viral titer until 14 dpi are referred to as Type B (Fig. 4). Additionally, we grouped organoids based on a morphological feature, to investigate the potential influence of choroid plexus structures, when present in addition to organoid tissue. This structure, as other regional identities, innately develops during organoids differentiation and can be identified on a macroscopic level due to the characteristic vacuole formation[23]. Additionally, the identity was confirmed using the

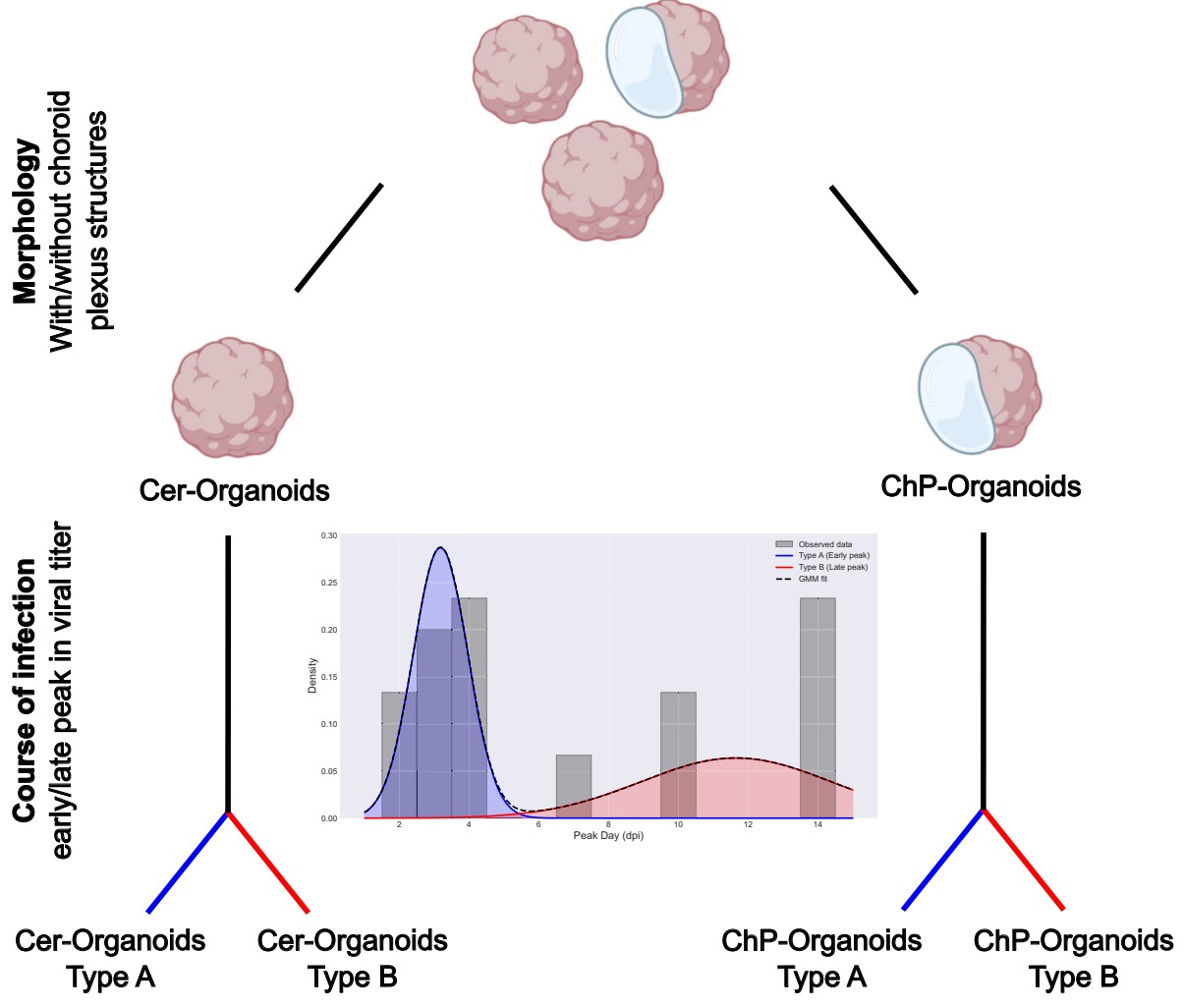

**Fig. 4 | Grouping of human WNV-infected human cerebral organoids.** As the infection of human cerebral organoids with WNV resulted in heterogeneous growth kinetics, subgrouping was performed based on two factors. First, their morphology, hence the presence (ChP) or absence (Cer) of choroid plexus structures. Second, the observed course of infection. The bimodal distribution of viral peak time points ($n = 30$) was analyzed using a Gaussian mixture model (GMM) with two components. Type A (Early peak): $\mu = 3.17$, $\sigma = 0.78$, weight=0.56. Type B (Late peak): $\mu = 11.65$, $\sigma = 2.72$, weight = 0.44. Due to biological reasons, two organoids were manually assigned to Type B (for details see Methods). The resulting groups include organoids that either reached their highest viral titer until 4 dpi (Type A;) or later until 14 dpi (Type B). Partially created in BioRender. Steffen, J. F. (2026) https://BioRender.com/vufhjff. Source data are provided as a Source Data file.

markers aquaporin-1 (AQP1) and chloride intracellular channel protein 6 (CLIC6) (Supplementary Fig. 1c). In the following, organoids with choroid plexus structures are referred to as ChP-organoids, and those lacking this structure are referred to as Cer-organoids.

Upon comparison of these groups in regards to WNV replication, more homogenous results in mainly Type A but also Type B ChP-organoids were observed in comparison to Cer-organoids (Fig. 5). Overall, a decrease in titer could be observed for Type A organoids upon reaching their peak until 4 dpi, while Type B organoids revealed a steady increase until their peak at 14 dpi at the latest. The previously described potential viral clearance could be observed for about 64 % of Type A organoids (3/5 Cer and 4/6 ChP). Conversely, no such effect was observed for Type B organoids until 14 dpi. When observing later time points about 15 % (1/7 Cer and 1/6 ChP) showed a decrease in titer below the LOD until 28 dpi. The remaining Type B organoids appeared to sustain a relatively long-term infection with WNV. This overall led to

the conclusion that the cerebral organoid model has the capacity to reflect diverse courses of WNV infection in vitro.

Furthermore, we observed differences in the neuroinflammatory profile when considering the course of infection as well as the presence of choroid plexus structures. Independent of the morphology, a trend for an increased release of CXCL10, CX3CL1, IL-1RA and sTREM-1 was observed for organoids with an early peak in viral titer until 4 dpi (Type A) when compared to organoids with a later peak until 14 dpi (Type B) (Fig. 6). Moreover, for some markers this effect was only observed in either ChP-organoids (with fluid-filled vacuole compartments) or Cer-organoids (without fluid-filled vacuole compartments). Hence, a trend for higher IL-6 and CCL2 release from Type A Cer-organoids in comparison to Type B organoids of the same morphology was observed, while no strong difference was seen between Type A and B ChP-organoids. Moreover, this trend for a higher release by Type A organoids was observed for CCL17, but specifically in ChP-organoids and

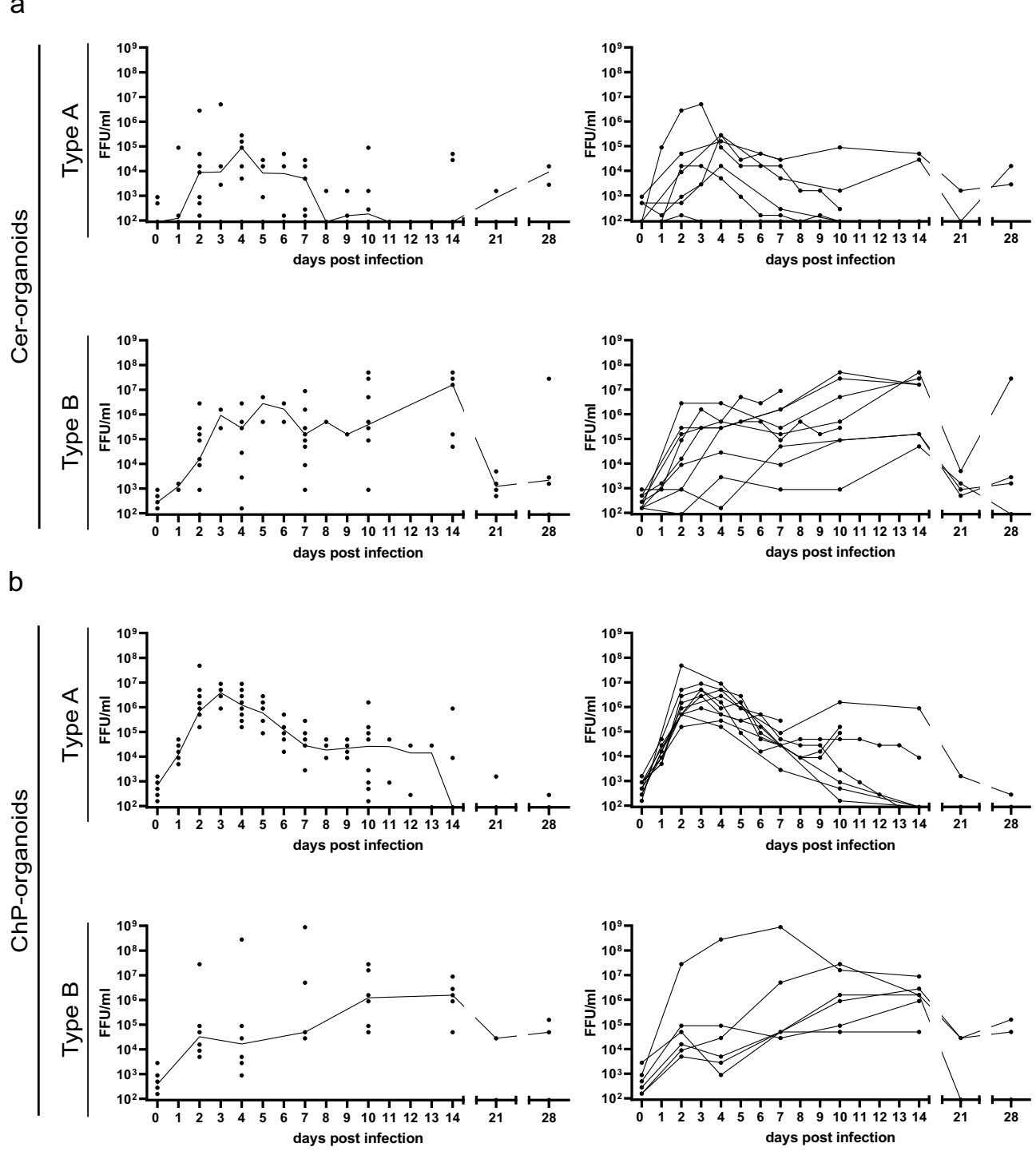

**Fig. 5 | Comparison of WNV replication in WNV-infected human cerebral organoids depending on their morphology and course of infection.** Human cerebral organoids were infected with $1 \times 10^4$ WNV infectious particles. Results were obtained in two independent experiments, both sampling to at least 14 dpi, whereby one experiment additionally included 21 and 28 dpi. Organoids were morphologically distinguished upon the **a** absence or **b** presence of choroid plexus (ChP) structures. In retrospect, organoids were furthermore grouped by the course of infection observed. Type A included organoids reaching their highest viral load until 4 dpi, whereas Type B organoids reached their highest viral load later than 4 dpi. The solid line in the graphs to the left is drawn through median values. The solid lines in the graphs to the right are drawn through the values of the individual organoids. Type A Cer-organoids: $n = 2–7$ per time point. Type B Cer-organoids: $n = 0–9$ per time point. Type A ChP-organoids: $n = 2–10$ per time point. Type B ChP-organoids: $n = 0–6$ per time point. Limit of detection (LOD): $88.5 \times 10^2$ FFU/ml. Source data are provided as a Source Data file. Exact sample sizes are provided in Source Data file.

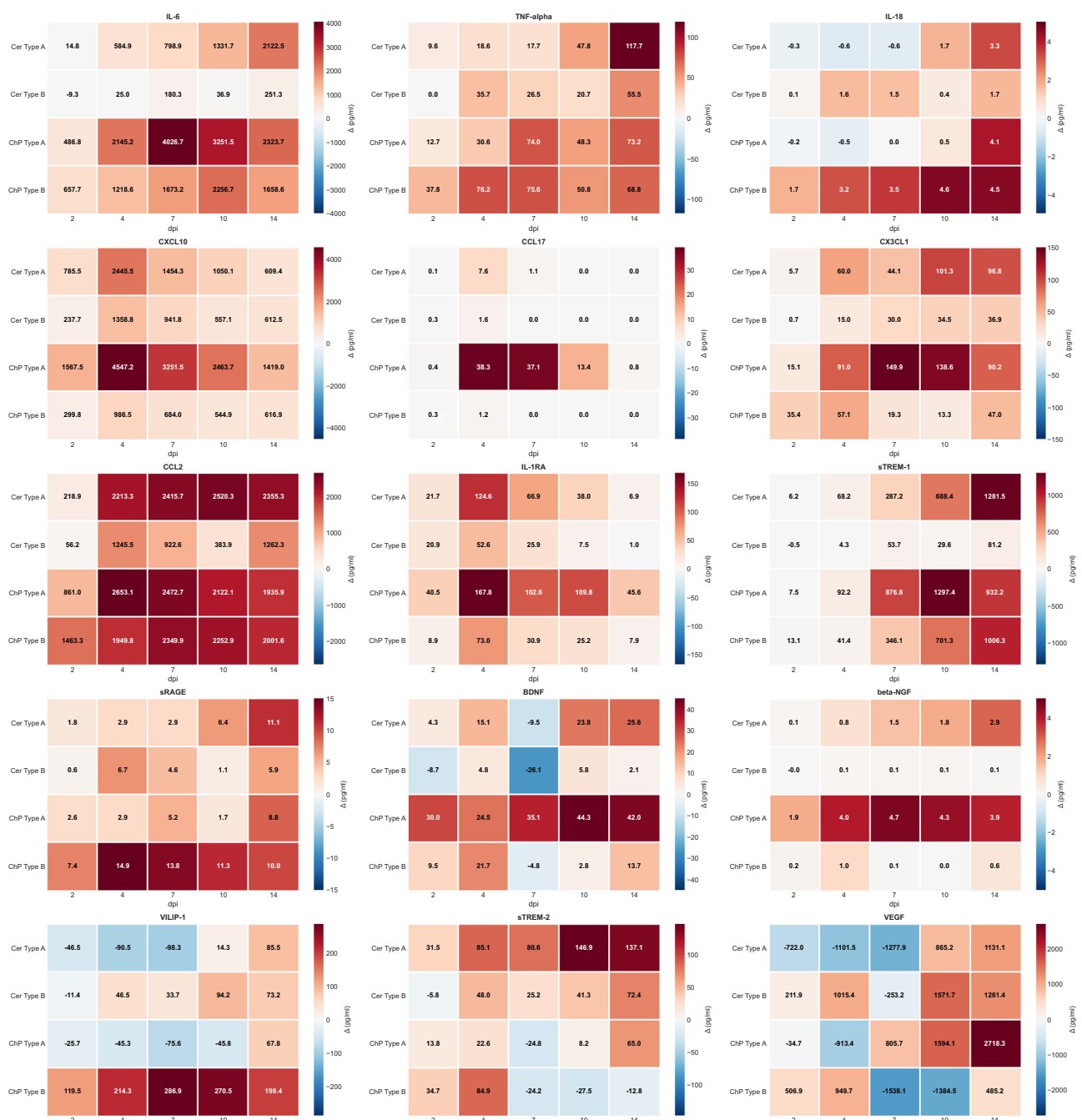

**Fig. 6 | Comparison of cytokines, chemokines and other biomarkers secreted by WNV-infected human cerebral organoids depending on their morphology and course of infection.** Screening of supernatants of WNV-infected human cerebral organoids for cytokines, chemokines and diverse biomarkers until 14 dpi using a bead-based LegendPlex™ assay. Organoids were subgrouped upon morphological identification of choroid plexus structures into choroid plexus (ChP) organoids, leaving all other regional identities that could not be identified by morphological features in the group without choroid plexus structures (Cer-organoid). Further subgrouping upon the course of WNV replication has Type A representing organoids with a peak in virus titer until 4 dpi, while Type B represents organoids with a later peak in virus titer until 14 dpi. Data are shown as Δ (infected − respective control) based on median absolute values. C-C motif chemokine 17

(CCL17). C-C motif chemokine 2 (CCL2). C-X3-C motif chemokine 1 (CX3CL1). C-X-C motif chemokine 10 (CXCL10). Beta nerve growth factor (β-NGF). Brain-derived neurotropic factor (BDNF). Interleukin 1 receptor antagonist protein (IL-1RA). Interleukin 6 (IL-6). Interleukin 18 (IL-18). Soluble receptor for advanced glycosylation end products (sRAGE). Soluble triggering receptor expressed on myeloid cells 1 (sTREM-1). Soluble triggering receptor expressed on myeloid cells 2 (sTREM-2). Tumor necrosis factor alpha (TNF-α). Vascular endothelial growth factor (VEGF). Visinin-like protein 1 (VILIP-1). Cer-organoids control: $n$ = 12–16 per time point. ChP-organoids control: $n$ = 11–15 per time point. Type A Cer-organoids: $n$ = 5–7 per time point. Type B Cer-organoids: $n$ = 7–9 per time point. Type A ChP-organoids: $n$ = 6–10 per time point. Type B ChP-organoids: $n$ = 6 per time point. Source data are provided as a Source Data file. Exact sample sizes are provided in Source Data file.

not Cer-organoids. Furthermore, for IL-18 the overall opposite trend for an increased release by Type B organoids was observed independent of morphology, while this trend was also seen for sRAGE but solely in ChP- and not Cer-organoids. Interestingly, the release of TNF-α revealed comparable levels in all subgroups, whereby only Type B ChP-organoids retained a statistically significant increase at 14 dpi (Supplementary Fig. 6).

Comparing organoids of different morphology (Cer-organoids/ChP-organoids) but with the same course of infection, revealed a trend towards higher marker levels in ChP-organoids for IL-6, CXCL10, CX3CL1, sTREM-1, and β-NGF independent of the course of infection, whereby this trend was only observed for Type A but not Type B for CXCL10 (Supplementary Figs. 6–9). Interestingly, the increased release of CCL17 was mainly attributed to only one group, ChP-organoids of the Type A, with only minor contribution of Type A Cer-organoids and none of Type B organoids of either morphology (Fig. 6).

Strikingly, besides revealing the trends in the release of biomarkers, grouping of the organoids revealed effects that could not be detected before. Hence, the trend for an increased release of β-NGF was observed in Type A organoids independent of the presence of choroid plexus structures. Furthermore, this trend was seen for sTREM-2 in Cer-organoids and for VEGF in ChP-organoids. The opposing trend for an increased release in Type B organoids was observed for VILIP-1 in ChP-organoids. For detailed information on the biomarker release upon organoid-subgrouping, see Supplementary Fig. 6–9.

## Discussion

Human WNV encephalitis, a severe and emerging infectious disease of global interest[4,5,24,25], is currently lacking a protective vaccine and targeted treatment options. Although studies on human patients are limited, neuronal damage as well as an involvement of astrocytes and microglia were demonstrated[14–17]. Furthermore, data obtained from animal models and in vitro experiments not only underlined the potential crucial role of astrocytes and microglia in the immune response, but also suggested their contribution to disease pathology[6,22]. In this study we describe the successful establishment of a human cerebral organoid WNV encephalitis model which provided insight into disease pathophysiology in a complex 3D environment, including astrocytes and microglia as innate immunocompetent cells.

We showed that the infection of 100-day-old human cerebral organoids with a low dose of WNV infectious particles resulted in robust infection. We observed diverse courses of infection, including a potential viral clearance in some organoids, while others sustained long-term infection for 28 days. Besides the acute clinical picture of WNV encephalitis, patients have been reported to suffer from long-term sequelae and the respective pathophysiology remains to be uncovered[6]. No persistent WNV infection of the CNS was reported for human patients as of now. Importantly, persistent WNV infection of brain tissue was reported in diverse animal models including rhesus monkeys and rodents[26,27]. Hence, it cannot be excluded that sequelae of WNV encephalitis might not only result from late consequences of acute inflammation and thereby inflicted damage, but also from persistent local infection with long-term inflammation. In both scenarios, human cerebral organoids appear well-suited for modeling acute WNV infection as well as WNV persistence in humans.

To further investigate potential targets of WNV in human cerebral organoids, immunofluorescent staining revealed perinuclear WNV envelope protein in small foci located at the outer cortical-like layers of the organoids, suggesting that a small number of WNV-infected cells can support the high viral titers observed. Interestingly, these areas were abundant in neurons, often observed as the main target of WNV in the CNS, but also astrocytes, which despite being reported less susceptible, were previously reported to support WNV infection[17,28]. Microglia on the other hand could not be observed in those areas

positive for the viral protein, but were distributed throughout the organoids. This might indicate that these cells are not preferentially attracted to the sites of infection. However, individual localization of microglia to infection foci might have stayed undetected, as the method is restricted to only small sections of the organoid. This is supported by the fact that we found chemoattractants of microglia, like CXCL10[29], in the supernatant of WNV-infected organoids. Hence, the attraction and migration of microglia is of interest for further studies, preferably by staining multiple cell layers or even whole organoids.

To gain insight into the response of human cerebral organoids to the infection with WNV, supernatants were screened for various cytokines, chemokines and other biomarkers which are important in the context of neuroinflammation, including astrocyte and microglia activation and communication. It must be noted that the effects observed in this study might result from active infection, as well as the exposure to inactive viral particles. Moreover, the response of uninfected bystander populations to released signaling molecules might contribute to the observed marker release. A statistically significant increase was observed in the early phase of infection for the pro-inflammatory cytokines IL-6, TNF-α and IL-18, the chemokines CXCL10, CCL17, CX3CL1, and CCL2, along with the biomarkers IL-1RA, sTREM-1, sRAGE, and BDNF. These findings highlight the capacity of the model to elicit an innate immune reaction and its advantage to enable the investigation of a biomarker responses in a time-dependent manner, capturing overall increased markers, like CXCL10, but also those increasingly released during a short and distinct period of the infection, as observed for IL-1RA. Especially such temporary effects will be hard to detect in other model systems, as the collection of CSF in animal models is an invasive procedure and therefore limited in the time intervals and sample volume when sampling one individual.

Notably, the screening method used in this study does not allow for assigning the released cytokines to a specific cell type, which will be of interest for future research. Nevertheless, several of the analytes that were observed to be increasingly released by infected organoids were previously reported to be involved in WNV encephalitis and/or astrocyte-microglia interaction. In this regard, IL-6 was reported to be released by primary microglia upon WNV infection[28], while astrocyte-derived IL-6 is discussed to promote pro-inflammatory functions of microglia[30]. Furthermore, the increased release of TNF-α observed in this study was previously reported in primary microglia upon infection with WNV[28], suggesting that TNF-α in this study might be at least partially produced by this cell type. Interestingly, microglia-derived TNF-α was furthermore reported to support a neurotoxic phenotype in astrocytes[31], highlighting its relevance for future studies on astrocyte-microglia interactions in WNV encephalitis. As microglia were not observed directly at WNV infection foci in this model, their involvement and functionality are interesting factors to be addressed in the future.

Moreover, the increased release of IL-18 observed in human cerebral organoids in this study was also shown for astrocytes and microglia infected with Japanese encephalitis virus, a close flavivirus relative to WNV. The capacity to subsequently promote the release of other pro-inflammatory cytokines by exactly these same cell types was shown, possibly resulting in a dysregulated loop with cytotoxic potential[32]. Thus, IL-18 might play a role in the activation of glia cells during WNV encephalitis, potentially contributing to pathology, which should be addressed in the future.

As described above, this study also found several chemokines to be released by human cerebral organoids upon infection with WNV. These factors are known chemoattractants for various immune cell types from the periphery[33–36]. As these cells are not included in our model, effects such as the potential viral clearance observed in the organoids seem to be achieved solely by the local glia cell population. Interestingly, CXCL10, a chemokine shown to be elevated in our

model, might cause neurotoxic effects[37]. In a mouse model, the deficiency of CXCR3, the receptor for CXCL10 on neurons, prevented apoptosis in primary neurons upon WNV infection. In this regard, TNF-α, a cytokine found to be increasingly released upon WNV infection in our model, was suggested to downregulate the CXCR3 expression, providing neuroprotection against elevated CXCL10 levels[37]. Indeed, despite the reports on neuronal damage and CPE in vitro using monocultures[17,38], there was no exuberant CPE observed upon WNV infection of human cerebral organoids in this study. This does not fully exclude cellular damage, but underscores that within this complex and partially immunocompetent in vitro system neuroprotective effects might limit neuronal damage. Of interest for further research in this context is also CCR2, the receptor for CCL2 previously reported to be found on neurons, astrocytes and microglia[39].

Besides the complex interplay of various immune responses, a straightforward strategy of immune regulation is the direct ligand-receptor binding by competitors. In this regard, either agonists bind to a relevant receptor without inducing a productive signaling, as reported for IL-1RA[40], or soluble receptor isoforms bind their ligands, subsequently preventing the binding to the respective membrane-bound receptor and thereby signaling[41]. Therefore, the increased release of IL-1RA, sTREM-1, and sRAGE observed in this study could play a role in immunoregulation in WNV encephalitis.

All of those effects of WNV infection on the release of soluble markers previously described were evaluated beyond the early phase of infection. Hence, long-term elevated levels of the pro-inflammatory cytokines IL-6, IL-18, and the chemokines CXCL10 and CX3CL1, and additionally sTREM-1 and BDNF, could be observed in this study. These findings again highlight the capacity of this model for studying not only the early response but also long-term effects of WNV encephalitis, including consequences of a persistent infection as well as post-infectious inflammation upon viral clearance, as the cause of WNV sequelae remains to be uncovered[6].

As previously described, we observed heterogeneous courses of WNV infection, reaching from potential viral clearance to long-term infection. To investigate whether the course of infection associates with patterns in the neuroinflammatory profile, we grouped our data depending on an early (Type A) or late (Type B) peak in viral titer. Notably, in about 64% of organoids with an early peak in viral titer (Type A) we observed a subsequent decrease in the release of infectious viral particles below the LOD during the early phase, hinting towards a viral clearance. Such an effect was not observed for organoids with a later peak in viral titer (Type B) during the early phase, with only about 15% reaching potential clearance in the late phase. The remaining organoids sustained a relatively long-term infection with WNV. Furthermore, organoids were grouped on the presence (ChP) or absence (Cer) of choroid plexus structures. The choroid plexus, forming the blood-CSF barrier of the brain, is, despite its importance in the CNS, often neglected. Besides its important functions in CNS homeostasis, it has been described to play various role in neurological infections, including the release of pro- inflammatory cytokines[42].

Upon subgrouping of the organoids, as described above, we discovered a trend towards a stronger increase of CXCL10, IL-1RA and sTREM-1 in organoids an early peak in viral titer (Type A), in comparison to those with a later viral peak (Type B), independent of choroid plexus presence. Such an increased response might contribute to the putative viral clearance characteristic of this group. Interestingly, the opposing trend became visible for IL-18. Hence, this factor might be of interest for long-term and potential persistent WNV-infection.

Interestingly, a trend was observed hinting towards a stronger increase of several markers, namely IL-6, CXCL10, CX3CL1, sTREM-1 and β-NGF in WNV-infected ChP-organoids in comparison to the morphologically broader group of Cer-organoids. ChP-organoids have been shown to develop epithelial cells of the blood-CSF barrier[43], which besides the production and secretion of CSF can also induce

proinflammatory signaling[42]. Hence, the previously described trend might be contributed to the specialized cells of the blood-CSF barrier of ChP-organoids. Additionally, subgrouping revealed previously hidden effects for β-NGF and VILIP-1. Consequently, considering factors like the morphology and course of infection revealed partially differing effects in the innate response of WNV-infected human cerebral organoids, highlighting the advantage of working with the heterogeneous model we present here but also region-specific organoids for follow-up research. Moreover, future experiments with increased sample sizes might allow for the detection of more factors for subgrouping and further analysis on observed effects, such as the potential viral clearance we observed in 64 % of organoids with an early peak in viral titer. Additionally, an increase in organoid batches would be beneficial for future research on this model, as the model's heterogeneity brings challenges to the data interpretation.

Concludingly, we demonstrate the value and usefulness of human cerebral organoids as a model to study WNV encephalitis. This complex 3D in vitro system allows insights in acute and persistent viral infections, and is suitable to study long-term consequences of the disease. The interplay of astrocytes and microglia, important innate immunocompetent cell populations of the CNS, and their contribution to viral defense and pathology, can be analyzed in detail. Moreover, in the future, the model can be further developed to investigate not only the innate but also the adaptive immune response by the addition of adaptive immune cells like T cells.

## Methods

### Human cerebral organoid generation and culture

Human cerebral organoids were generated from HMGU1 human iPSC, originating from fibroblasts[44] (male; provided by the Institute of Stem Cell Research (ISF) at Helmholtz Centre in Munich (HMGU), Germany (#1 cell line; hPSCreg ISFi001-A) using the Stemdiff™ Cerebral Organoids Kit (STEMCELL Technologies Germany GmbH, Cologne, Germany) and subsequently matured using the STEMdiff™ Cerebral Organoid Maturation Kit (STEMCELL Technologies Germany GmbH, Cologne, Germany), following the manufacturer's instructions. Briefly, stem cells were expanded in iPS Brew XF medium (Miltenyi Biotec, Bergisch-Gladbach, Germany) supplemented with 10 μM rho-associated protein kinase (ROCK) inhibitor (Selleckchem, Houston, TX, USA) for the first 24 hours on Matrigel® (Corning Inc., Glendale, AZ, USA) coated Nunc cell culture ware (Thermo Fisher Scientific, Waltham, MA, USA) until 70% confluency was reached and large, flat-edged stem cell cultures were formed. iPSC cultures were harvested and dispensed into a single cell solution using gentle cell dissociation reagent (STEMCELL Technologies, Vancouver, Canada) and 9,000 cells were seeded per well on a 96-well round-bottom ultra-low adhesion plate (Corning Inc., Glendale, AZ, USA) using seeding medium containing 4:1 basal medium 1 and supplement A, as well as 10 μM ROCK inhibitor. The forming embryoid bodies were fed at 48 h and 96 h after seeding with the same medium but without ROCK inhibitor. Neuronal induction was performed 5 days after embryoid body formation. Thus, embryoid bodies were transferred into 24-well ultra-low adhesion plates (Corning Inc., Glendale, AZ, USA) containing 99:1 parts of basal medium 1 and supplement B. After two days, organoids with round, optically translucent edges were induced in Matrigel®. For this, single embryoid bodies were covered with 15 μl of ice-cold Matrigel® and polymerization was performed for 30 min at 37 °C. Subsequently, Matrigel®-embedded organoids were washed into six-well ultra-low adhesion plates (Corning Inc., Glendale, AZ, USA) containing expansion medium consisting of 97:1:2 parts of basal medium 2, supplement C, and supplement D. After 72 h, organoids with budding neuroepithelium on their surface formed. From this point on, medium was exchanged every three to four days using organoid maturation medium (STEMCELL Technologies) containing 98% basal medium 2 and 2% supplement E. Culture vessels were kept on an orbital shaker (Vevor

Corporation GmbH, Cologne, Germany) at 60 rpm, 37 °C, and 5% $CO_2$ until organoids were used 100 days after generation. Organoids were considered mature at day 40, however, to ensure glia formation they were kept under the previously described conditions until day 100.

## Virus amplification and titration

The WNV strain used (clade 1a, strain TOS-09, GenBank HM991273/ HM641225) was amplified on Vero cells (*Chlorocebus sabaeus*; CVCL_0059, obtained from ATCC, Cat# CCL-81) kept in Dulbecco's modified Eagle Medium (DMEM) (PAN-Biotech GmbH, Aidenbach, Germany) supplemented with 5% FCS (Biochrom AG, Berlin, Germany) and 1% penicillin/streptomycin (p/s). Virus stock was added into the medium and cells were incubated at 37 °C and 5% $CO_2$ until a pronounced cytopathic effect (CPE) was observed. The supernatant was centrifuged at 1000 x g for 5 min and subsequently polyethylene glycol (PEG)-precipitation was performed using the PEG Virus Precipitation Kit (Abcam, Cambridge, UK) following the manufacturer's instructions. Virus stocks were stored at −80 °C. Titers were quantified via TCID50 assay in serial 10-fold dilution steps in DMEM culture medium (PAN-Biotech GmbH, Aidenbach, Germany) on Vero cells (*Chlorocebus sabaeus*; CVCL_0059, obtained from ATCC, Cat# CCL-81). The CPE was evaluated 7 days post-infection and the viral titers were calculated based on the Spearman-Kaerber method[45].

## WNV infection of human cerebral organoids

Infection experiments were conducted under biosafety level 3 conditions in two independent experiments with two organoid batches to evaluate the early phase of infection until 14 days post infection (dpi). Later time points of late phase, 21 and 28 dpi, were obtained in one experiment and hence one batch.

Infection was performed with 100-days-old cerebral organoids by adding 500 μl fresh maturation medium containing $1 \times 10^4$ WNV focus forming units (FFU), each cultured in an individual well. In total, 20 organoids were infected in experiment 1 and 27 organoids in experiment 2. The negative controls received a comparable treatment with the respective volume of PEG-precipitated Vero cell culture supernatant. Organoids with a diameter of 2 – 3 mm and an average cell number of roughly $3 \times 10^6$ imply an estimated multiplicity of infection (MOI) of approximately 0.003. After 1 h incubation at 37 °C and 5% $CO_2$ the infection medium was discarded, fresh maturation medium was added and t0 samples were taken. Thereafter, supernatants were sampled at the following time points:

Experiment 1: Daily until 14 dpi. Sample sizes per time interval: $n = 20$ (0 − 2 dpi); $n = 16$ (3 − 4 dpi); $n = 12$ (5 − 7 dpi); $n = 8$ (8 − 10 dpi); $n = 4$ (11 − 14 dpi).

Experiment 2: Sampling on 2, 4, 7, 10, and 14 dpi and additionally at 21 and 28 dpi. Sample sizes per time point: $n = 27$ (0, 2 dpi); $n = 20$ (4, 7, 10, 14 dpi); $n = 12$ (21, 28 dpi). To ensure comparability of both experiments, for the days where no sample is taken until 14 dpi, medium is simply discarded and replaced.

The sampled volume was replaced by fresh maturation medium and samples were centrifuged at 2000 x g for 2 min and subsequently stored at −80 °C. Additionally, organoids were harvested on the following days:

Experiment 1: Harvest on 2, 4, 7, 10, and 14 dpi. Sample sizes per time point: $n = 4$.

Experiment 2: Harvest on 2, 14 and 28 dpi. Samples size per time point: $n = 7$ (2 dpi), $n = 8$ (14 dpi), $n = 12$ (28 dpi).

Briefly, organoids were transferred into 2 ml safelock reaction tubes (Sarstedt AG & Co. KG, Nümbrecht, Germany) with 1.8 ml buffered 4% formaldehyde solution (DIAPATH, Martinengo, Italy). The solution was replaced after 24 h, followed by an incubation period of another 24 h. Fixed organoids were then transferred onto parafilm (Bemis, Oshkosh, NE, USA) without fluid. 30 μl of 2% agar (Carl Roth GmbH+Co. KG, Karlsruhe, Germany) were added and once solidified,

the embedded organoids were transferred into histology cassettes containing filter paper, placed into glass cuvettes and placed under running water. After standard dehydration processing with ethanol (Carl Roth GmbH+Co. KG, Karlsruhe, Germany), xylene (Carl Roth GmbH + Co. KG, Karlsruhe, Germany) and paraffin (SLEE medical GmbH, Mainz, Germany), organoids were embedded into paraffin blocks, cut into 2 μm sections using a microtome (Leica Instruments GmbH, Nussloch, Germany), and finally transferred onto adhesive microscopy slides (Paul Marienfeld GmbH & Co. KG, Lauda-Königshofen, Germany).

## Viral growth kinetics and organoid subgroup analysis based on course of infection and morphological characteristics

Infectious particles released by WNV-infected organoids into the supernatant were quantified via TCID50 assay as previously described for viral stocks. The data was analyzed for the distribution of days viral peak titers were reached in the course of infection of individual organoids. Organoids that were taken out of the experiment before 7 dpi could not be assigned, which lead to their exclusion for this dataset. Two organoids were excluded from this analysis as they did not reach one clear peak but reached their highest viral titer at several peaks. One organoid reached the first viral peak at 5 dpi. Importantly, this peak was stable until 6 dpi and upon a decrease in titer at 7 dpi, peak titer was again reached at 8 dpi. One organoid showed one peak at 2 dpi and the second peak was reached at 7 dpi and was furthermore stable for measurements at 10 and 14 dpi. As the resulting bimodal distribution of peak days indicated two different groups, a Gaussian mixture model (GMM) with two components was applied. The GMM was fitted to the distribution of peak days. For reproducibility the model was initialized with random_state=42. The excluded organoids were manually assigned to Type B as their course of infection is more closely represented by these organoids. Subsequently, organoids with an early peak in viral titer until 4 dpi are further referred to as Type A, while organoids with a late peak in viral titer until 14 dpi are referred to as Type B. This analysis was performed using Python (version 3.13.5) with the following packages: pandas (version 2.3.1), numpy (version 2.2.6), sklearn (version 1.6.1), spicy (version 1.16.0), matplotlib (version 3.10.3), seaborn (0.13.2), openpxyl (version 3.1.5). Python code was developed with the support of Claude (Anthropic, Sonnet 4.5) and is based on established packages with no custom analysis. All code outputs were manually reviewed and validated.

Additionally, organoids were grouped by the presence or absence of choroid plexus structures, which were identified on a macroscopic level as fluid-filled vacuole compartments[43]. In summary, the organoids were grouped into four groups, namely Cer-organoids (no vacuoles) Type A (peak titer 4 dpi) and Type B (peak titer between 4 and 14 dpi) and ChP-organoids (vacuoles) Type A and Type B. Clearance rates were calculated using organoids with a minimum of 14 dpi.

## Antigen specific staining of WNV and cells in cerebral organoids

The formalin-fixed paraffin-embedded (FFPE) samples of infected and uninfected organoids were stained for cellular markers including microtubule-associated protein 2 (MAP2) for neurons, glial fibrillary acidic protein (GFAP) for astrocytes, and macrosialin (CD68) for microglia, plus WNV envelope protein to investigate the viral cellular targets. Notably, cerebral organoids containing fluid-filled vacuoles, representing choroid plexus identities, were not included in this assay, as the vacuoles made up the majority of the sample, leading to most sections of the samples to consist of only vacuole membrane.

After deparaffinization, antigen retrieval was performed by cooking of the samples for 2 min in citrate buffer using a pressurized cooker. Subsequently, three washing steps were performed, each for 5 minutes in PBS buffer on a shaker. This procedure was performed between all of the following steps. To block non-specific background, samples were incubated in 5% bovine serum albumin (BSA) in PBS

buffer for 30 min at room temperature. Next, primary antibody staining was performed using the following dilution of the stated antibodies in PBS buffer over night at room temperature: WNV envelope protein (Novusbio NBP3-13574; mouse) 1:300, MAP2 (Abcam ab183830; rabbit) 1:600, GFAP (Zytomed RBK037; rabbit) 1:100, CD68 (Cell Signaling #76437; rabbit) 1:100. The following day, the respective secondary antibody staining was performed for 2 h at room temperature: goat anti-rabbit IgG Alexa Fluor™ 568 (Invitrogen #A-11011), goat anti-mouse IgG Alexa Fluor™ 647 (Invitrogen #A-21235). A final washing step in aqua bidest. was performed to prevent crystallization of PBS salts. Lastly, slides were mounted with Roti®Mount FluorCare 4′,6-diamidin-2-phenylindol (DAPI) (Carl Roth GmbH + Co. KG, Karlsruhe, Germany). Samples were analyzed using the Olympus FV3000 confocal microscope (Olympus, Tokyo, Japan) and Olympus FV31S-SW software (version 2.6.1.243) and further processed in Fiji (ImageJ 1.54 g)[46]. Brightness and contrast were adjusted for optimal visualization using the "Brightness/Contrast" tool and "Unsharp Mask" tool.

For confirmation of cerebral organoid cell types two markers per cell type have been stained. Namely, MAP2 and RNA binding protein fox-1 homolog 3 (Neuronal nuclei antigen; NeuN) for neurons, GFAP and Protein S100-B (S100B) for astrocytes and CD68 and CX3C chemokine receptor 1 (CX3CR1) for microglia. For confirmation of choroid plexus structures aquaporin-1 (APQ1) and chloride intracellular channel protein 6 (CLIC6) have been stained. S100B (1:100; Antibodies Online ABIN2840160; rabbit) and CX3CR1 (1:100; Invitrogen #702321; rabbit) have been processed for immunofluorescence staining as described above. Sections were analyzed using the Zeiss Axio Imager M1 microscope (Zeiss, Oberkochen, Germany) and ZEISS Axio Imager software (v1.0.2.1) and further processed in Fiji (ImageJ 1.54 g)[46]. Brightness and contrast were adjusted for optimal visualization using the "Brightness/Contrast" tool. MAP2, NeuN, GFAP, CD68, APQ1 and CLIC6 have been processed for immunohistological staining. Briefly, upon deparaffinization and antigen retrieval endogenous peroxidase blocking was performed. Primary antibody staining was performed over night at room temperature: MAP2 (Abcam ab183830; rabbit) 1:300, NeuN (Cell Signaling 24307; rabbit) 1:100, GFAP (Zytomed RBK037; rabbit) 1:200, CD68 (Cell Signaling #76437; rabbit) 1:200, APQ1 (ThermoScietific MA5-32593; rabbit) 1:100 and CLIC6 (Thermo Scientific PA5-101519; rabbit) 1:100. This was followed by immuno-peroxidase staining using the DCS-AEC 2 component detection kit (DCS-diagnostics, Hamburg, Germany) and immunophosphatase staining using the DCS AP Detection Kit (DCS-diagnostics, Hamburg, Germany) and new fuchsine substrate. Lastly, standard haematoxylin staining was performed and slides were mounted using Kaiser's glycerol gelatine (Carl Roth GmbH + Co. KG, Karlsruhe, Germany). Samples were analyzed using the Zeiss Axiophot (Zeiss, Oberkochen, Germany) microscope and ZEN blue software (version 3.2.0.0000). Brightness and contrast were adjusted using Inkscape 1.3.2 (091e20e, 2023-11-25, custom).

### Cytokine measurements and comparative analysis of grouped data

Supernatant samples were analyzed for biomarker concentrations by LEGENDplex™ assays (BioLegend, Fell, Germany). LEGENDplex™ Human Neuroinflammation Panel 1 (13-plex; 740796) and LEGENDplex™ Human Macrophage/Microglia Panel 1 (13-plex; 740502) (BioLegend, San Diego, CA, USA) were used, providing the detection of the following analytes: arginase, brain-derived neurotrophic factor (BDNF), C-C motif chemokine 17 (CCL17), C-C motif chemokine 2 (CCL2), C-X3-C motif chemokine 1 (fractalkine; CX3CL1), C-X-C motif chemokine 10 (CXCL10), interferon gamma (IFN-γ), interleukin (IL-10), interleukin 12 subunit p40 (IL-12p40), interleukin 12 subunit p70 (IL-12p70), interleukin 18 (IL-18), interleukin 1 receptor antagonist protein (IL-1RA), interleukin 1 beta (IL-1β), interleukin 23 (IL-23), interleukin 4

(IL-4), interleukin 6 (IL-6), soluble receptor for advanced glycosylation end products (sRAGE), soluble triggering receptor expressed on myeloid cells 1 (sTREM-1), soluble triggering receptor expressed on myeloid cells 2 (sTREM-2), transforming growth factor beta 1 (TGF-β1), tumor necrosis factor alpha (TNF-α), vascular endothelial growth factor (VEGF), visinin-like protein 1 (VILIP-1), beta nerve growth factor (β-NGF). Samples from WNV-infected organoids were compared to those of uninfected organoids of the respective time points. Data analysis was performed using the LEGENDplex™ Qognit software provided by BioLegend. Samples were analyzed with the BD LSRIIFortessa™auto-sampler (HTS) (BD Biosciences, San Jose, CA, USA) using the BD FACSDiva Software (v8.0). Statistical analysis was performed with GraphPad Prism 9 (GraphPad Software Inc., La Jolla, CA). A mixed-effects model (REML) with Šídák's multiple comparisons was used for statistical analysis of differences between infected organoids and uninfected controls.

Analysis of differences in the dataset upon grouping based on the course of infection (Type A/B) and presence or absence of choroid plexus structure (Cer/ChP) was performed as described above. Additionally, to visualize trends, median values of all time points were used to calculate the delta of infected samples to their uninfected controls (Type A/B Cer to Cer controls; Type A/B ChP to ChP controls). This analysis was performed using Python (version 3.13.5) with the following packages: pandas (version 2.3.1), numpy (version 2.2.6), matplotlib (version 3.10.3), seaborn (0.13.2), open-pxyl (version 3.1.5). Python code was developed with the support of Claude (Anthropic, Sonnet 4.5) and is based on established packages with no custom analysis. All code outputs were manually reviewed and validated.

### Reporting summary

Further information on research design is available in the Nature Portfolio Reporting Summary linked to this article.

## Data availability

Source data are provided with this paper.

## Code availability

Code is available via Zenodo (https://doi.org/10.5281/zenodo. 18628798).

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

## Acknowledgements

This work was supported by the Joachim Herz Foundation, Hamburg as part of the graduate school "Infection Biology of Tropical Pathogens" at the Bernhard Nocht Institute for Tropical Medicine (grant number 34901609). Work by L.W. was supported by the German Federal Institute for Risk Assessment (BfR, 60-0102-01.P616), the German Center for Infection Research (DZIF, TTU EI) as well as the Collaborative Research Center (CRC-1648). We gratefully acknowledge Giada Rossini (Microbiology Unit, IRCCS Azienda Ospedaliero - Universitaria di Bologna, Italy) for providing the WNV strain. We thank Unchana Lange and Michelle Helms for their invaluable support in the cell culture and Petra Allartz (all BNITM, Hamburg) for her excellent technical assistance to the immunofluorescence staining. We gratefully acknowledge Jonas Schmidt-Chanasit (BNITM, Hamburg) for his contribution to the funding of this project.

## Author contributions

J.F.S. contributed to study design, conducted most of the experiments, analyzed the data and wrote the original manuscript draft. L.W. contributed to experiments. S.J. and D.T. supervised the project and contributed to study design. D.T. provided financial support. J.F.S., L.W., S.J., and D.T. corrected the manuscript draft.

## Funding

## Competing interests

The authors declare no conflicts of interest.
