## [Peer Review file · Nature Communications]

A human cerebral organoid model of West Nile virus encephalitis shows innate immunocompetency

Corresponding Author: Dr Johanna Steffen

Version 0:

Reviewer comments:

Reviewer #1

(Remarks to the Author)

In this manuscript, the authors use neural organoids to establish a model for West Nile Virus (WNV) encephalitis. This topic is highly relevant and novel as the use of human models for human viral infections is critical for a complete understand of human pathology. To date, this is the first study to focus on WNV using these models.

The authors use of two different models, cerebral organoid (Cer) and choroid plexus (ChP), and describe that upon infection of either model with WNV, there are two courses of replication kinetics: some organoids showing a peak in infectivity in the initial 4 days (Type A), while others show a peak in infectivity around day 14 (Type B). The authors then assess neurotropism in the Cer organoids, claiming that WNV particles are present in neuron and astrocytes regions devoid of microglia. Lastly, they analyse the secretion of a pre-selected panel of cytokines and chemokines and show different responses depending on organoid morphology and course of infection. Although the findings and research on this topic are of high interest, there are concerns regarding the experimental approach that make it difficult to assess the biological foundation of the described results.

There are several major points of concern in the current form of the manuscript, which I discuss below:

1. The authors describe that they use the commercial kit from STEMCELL Technologies to generate their cerebral organoids and obtain a mixed culture of Cer and ChP organoids. Is this inconsistency expected when using this protocol? What is the overall percentage of Cer vs ChP organoids in one batch? Have the authors confirmed that what they classify as ChP organoids indeed show ChP identity? Characterization of this model is necessary before further assumptions can be made. Lastly, why have the authors use the ChP organoids for infection? This should be justified in the manuscript.
2. It is generally accepted that microglia are not present in cerebral organoids with the exception of one report in the literature (Ormel et al. 2018). The authors show CD68 staining in the necrotic core of the organoid in WNV-infected organoids to demonstrate that microglia do not migrate to areas of infection. Can the authors provide characterization of the microglia in non-infected organoids? In this situation, microglia would be expected in the cortical regions containing neurons and astrocytes.
3. Please clarify in the methods section whether infection was performed in single organoids or whether multiple organoids were infected in the same well. Please specify how many organoids were infected per experiment. According to the methods section in experiment 2, time point 0 was not collected, but there are data points that seem to belong to experiment two (Figure 2). Please clarify.
4. The authors claim that the organoids naturally present different courses of disease. However, according to the methods section, medium was only changed on collection time points, indicating lack of consistency between experiments 1 and 2, which could influence the time point at which the peak in infectivity is reached. For example, it is noticeable that ChP type B organoids (Figure 2B) are all from experiment 2. Similarly, in the Cer Type B organoids, only 2 out of 9 are from experiment 1. How is this taken into consideration for the data analysis and interpretation of Figure 2? Further experiments using the exact same set-up would be necessary to establish that the difference between type A and type B responses is not due to the sampling strategy.
5. Another important factor that may affect the replication kinetics is the size of the organoids used for infection. The manuscript reports that organoids were infected with 1×10^4 FFU/organoid. Given the difficulty of establishing MOI for organoids due to their 3D and complex nature, it is important to select organoids of approximately the same size for infection. Have the authors taken this into consideration? Was there a size difference between type A and type B organoids? This will have an impact when comparing Cer with ChP samples, as by the description of the ChP in lines 175-177 of the

manuscript, the MOI is much higher in ChP vs Cer organoids. Further direct comparisons between the two models should take this into consideration and should also be mentioned in the discussion.

6. Figure 3 shows staining of WNV in neuron- and astrocyte-rich regions. Is this distribution the same for type A and type B? Please show data for both situations.

7. What is the purpose of Figure 4? It seems misleading to classify acute and post-acute phase when discussing type A and type B together since they have different courses of infection.

8. In Figure 5, are the levels normalized to the mock situation of each of the models? This would make it easier to evaluate differences.

9. Please provide the N of organoids used for analysis for each readout, either in the methods section or in the figure captions.

10. Lastly, the authors claim that the observation of two types of infection course is representative of the ability of the organoids to represent disease heterogeneity. However, given the lack of biological replicates (both at the host and viral strains levels), it is difficult to support this conclusion.

Reviewer #2

(Remarks to the Author)

Reviewer #3

(Remarks to the Author)

The manuscript "A human cerebral organoid model of West Nile virus encephalitis shows innate immunocompetency" by Steffen et al. describes the use of human cerebral organoids for studying WNV induced encephalitis. This is an interesting model that takes into account the complex 3D microenvironment in the brain. Overall the manuscript is well written with only minor editing for the English language required. Over all, the data support the conclusions however some of the subgrouping of data seems artificial as described below in the comments.

Major comments

- Line 225, see comment line 162 below. The kinetics shown in figure 1 and used for grouping, do not represent the kinetics observed in figure 2, at least not for B. Instead the "B group" organoids seem to replicate as efficiently as A, initially, but while A organoids show transient infection, the B group organoids remain infected for at least 14 days. Unless the authors have a clear biological explanation for the difference between the 2 "groups", this distinction seems to be artificial. Therefore the analyses presented starting at line 294, are difficult to interpret.

-

Minor comment

- Line 78, while the use of 2 different batches controls for some of the biological variability, results are still based on cells from a single donor?

- Line 128, although determining the multiplicity of infection with a complex cells system such as an organoid is difficult, it would be good to indicate whether the dose used, would be high or low, perhaps by providing an average number of cells in organoids.

- Line 162-164. The rationale for grouping organized based on their replication kinetics is not clear. Does this suggest different stages of development, or is it merely a result of the variability?

- Line 221, see previous comment line 128, why is this considered a low dose, compared to what?

- Line 259. Staining for MAP2 and GFAP does not seem to overlap with WNV staining, so why are neurons and astrocytes potential targets? Did you attempt to stain for multiple markers and virus in the same organoid, so MAP2, GFAP AND virus?

- Line 334, since there are no immune cells involved in WNV infection in this model, either because they are absent, or do not react, can you state that this is inflammation? Or merely the release of proinflammatory cytokines and chemokines, which in vivo would lead to inflammation?

- Line 433, again the subgrouping and finding of different trends is highlighted, but a discussion on what the biological relevance of these subgroups A and B is lacking.

Version 1:

Reviewer comments:

Reviewer #1

(Remarks to the Author)

The authors have positively addressed most of the concerns raised and improved the quality of the manuscript. In my opinion the main concern at this stage is the biological significance of these results in the context of WNV infection of the CNS. The manuscript mainly reinforces the use of neural organoids to investigate viral infections and highlights variability in replication kinetics and immune response the WNV but does not address underlying mechanisms. Additional points of concern are outlined below:

1- In Figure 3, the authors show and describe in length an inflammatory profile over time upon WNV infection. It is not clear why they chose to do so as it represents all the organoids used in the study which the authors later subclassify as different courses of disease showing distinct profiles. For this reason, I find it misleading to describe a unique profile when this is not the case and suggest that the authors remove this figure from the manuscript.

2- It should also be noted, with regards to the inflammatory profile, that the authors did not include a heat inactivated control on these experiments and therefore should discuss the possibility that some of the observed responses, especially acute, may be due to exposure to inactive viral particles and not to active infection. This should be discussed.

3- In line 153 the authors provide the rationale for classification of type A and type B organoids based on the timepoint of peak infectivity. Why is this the only factor considered? In line 176, the authors state that 64% of type A organoids show viral clearing. Why are these not considered a third course of infectivity and constitute a third group? Do these organoids show a distinct immune profile?

4- The authors should address in the discussion the challenges that the model heterogeneity brings to the data interpretation, especially given the low ($n=2$) number of batches used.

5- In lines 235-241 of the discussion authors discuss the WNV neurotropism suggesting that neurons and astrocytes are the targets. However, the authors do not demonstrate colocalization of WNV with any of the used cell markers, and therefore fail to show neurotropism, this should be clear in the discussion. Further in lines 241-243, authors suggest that the lack of microglia in the site of infection suggests that they are not directly involved in the response to WNV. Is it possible that the microglia are not functional? This should be addressed.

Reviewer #2

(Remarks to the Author)

Reviewer #3

(Remarks to the Author)

the authors have adequately addressed this reviewers comments

Rebuttal letter

Dear Editor and Reviewers,

We would like to thank you for the thorough review of our manuscript. We appreciate your valuable comments and suggestions, which helped us to improve clarity and quality of our manuscript. Firstly, we adjusted the order of our presented data for improved clarity, showing results for human cerebral organoids replication, immunofluorescence and temporal biomarker release and then present the analysis upon grouping the infected organoids. Moreover, we now provide additional characterization of our model, by verifying makers for microglia as well as choroid plexus and demonstrate the distribution of cell types in the absence of WNV. Furthermore, we elaborate more on the rationale of analyzing our heterogenous data by subgrouping based on the course of infection and the presence or absence of choroid plexus structures. Additionally, we provide a more advanced mathematical approach for the distinction of two types of replications and improved the visualization of their influence on the neuroinflammatory profile.

Below we address each individual comment point-by-point. The reviewers' comments are reproduced in bold, followed by our responses in plain text. All changes in the revised manuscript are highlighted in blue.

Reviewer #1

1. The authors describe that they use the commercial kit from STEMCELL Technologies to generate their cerebral organoids and obtain a mixed culture of Cer and ChP organoids. Is this inconsistency expected when using this protocol? What is the overall percentage of Cer vs ChP organoids in one batch? Have the authors confirmed that what they classify as ChP organoids indeed show ChP identity? Characterization of this model is necessary before further assumptions can be made. Lastly, why have the authors use the ChP organoids for infection? This should be justified in the manuscript.

> Indeed, the formation of various regional identities using the STEMCELL Technologies kit is to be expected. Unlike protocols or commercial kits aiming for region-specific organoids, the Kit is based on the original Lancaster publication leading to the formation of heterogenous cerebral organoids, containing various regional identities, including those representing choroid plexus (Lancaster et al. 2013, Nature, PMID: 23995685); Lancaster, Knoblich, 2014, Nat Protoc. PMID: 25188634). Hence, the subgrouping we performed to further analyze our data on the presence (ChP) or absence of choroid plexus structures (Cer) does not describe two completely different models, but the same model that due to heterogenous differentiation either did or did not present with choroid plexus structures.

> The percentage of organoid presenting with choroid plexus structures was relatively consistent and around 30% each batch.

> Besides the characteristic vacuole formation (Pellegrini et al., 2020, Science, PMID: 32527923) that can be evaluated on a macroscopic level, we confirmed the choroid plexus identity by immunohistology staining of the choroid plexus specific markers AQP1 and CLIC6 (Supplementary Figure 1). This is consistent with previous publications on choroid plexus organoids (Pellegrini et al., 2020, Science, PMID: 32527923).

> Our study is based on a pioneer approach, hence the cerebral organoids containing various identities were chosen as a model. As described above, choroid plexus identities are one of

many regional identities of the brain that can form in human cerebral organoids and are easily identifiable on a macroscopic level, as they present vacuole structures. Moreover, grouping upon the presence or absence of choroid plexus structure is of interest for our analysis on the neuroinflammatory profile as ChP-organoids have been shown to develop epithelial cells of the blood-CSF barrier (Pellegrini et al., 2020, Science, PMID: 32527923), which besides the production and secretion of CSF can also induce proinflammatory signaling (Thompson et al. 2022, Fluid Barriers, PMID: 36088417) (Lines 322-325, 334-336). Indeed, we observed trends for a stronger increase of several biomarkers (IL-6, CXCL10, CX3CL1, sTREM-1 and b-NGF) in organoids with choroid plexus structures (ChP) in comparison to those without (Cer), which therefore might be of interest for future research.

2. It is generally accepted that microglia are not present in cerebral organoids with the exception of one report in the literature (Ormel et al. 2018). The authors show CD68 staining in the necrotic core of the organoid in WNV-infected organoids to demonstrate that microglia do not migrate to areas of infection. Can the authors provide characterization of the microglia in non-infected organoids? In this situation, microglia would be expected in the cortical regions containing neurons and astrocytes.

> To date, there are different approaches described to generate cerebral organoids. The Kit by STEMCELL Technologies is using an unguided approach based on the Lancaster publication. As this approach does not use Dual-SMAD inhibition (Lancaster et al. 2013, Nature, PMID: 23995685), it does not only allow for ectoderm to arise and build neuroectoderm, leading to cerebral organoids containing neuronal cell types like neurons and astrocytes, but also the formation of mesoderm (Bershteyn, Kriegstein, 2013, Cell, PMID: 24074857). This is in line with findings by Ormel et al. showing the innate formation of the mesoderm-derived microglia in organoids based on protocols without Dual-SMAD inhibition. In addition to CD68 we verified the microglia-specific marker CX3CR1 in our cerebral organoids, ensuring their microglia identity (Supplementary Figure 1). Overall, we observed microglia to be distributed throughout the whole organoids in a heterogenous manner, including outer cortical layers as well as less organized inner regions. However, we were not able to detect them at infection foci. The limitations of our staining method are described in the manuscript (Lines 243-248), as we cannot fully exclude microglia migration or infection. In addition to our immunofluorescence staining, we added immunohistochemical staining of the important cell markers in an uninfected organoid as well as an infected organoid to the supplementary data (Supplementary Figure 1), showing the distribution of cells. Here, microglia were also shown in outer layers of the organoid.

3. Please clarify in the methods section whether infection was performed in single organoids or whether multiple organoids were infected in the same well. Please specify how many organoids were infected per experiment. According to the methods section in experiment 2, time point 0 was not collected, but there are data points that seem to belong to experiment two (Figure 2). Please clarify.

>We thank the reviewer for pointing this out. We now clarified in the methods section that infection was performed in single organoids in one well and added the number of infected organoids per experiment (Lines 405-407). Indeed, time point 0 was taken for the analysis of viral titers in both experiments, which is described in the methods section (Lines 411-412).

4. The authors claim that the organoids naturally present different courses of disease. However, according to the methods section, medium was only changed on collection time points, indicating lack of consistency between experiments 1 and 2, which could influence the time point at which the peak in infectivity is reached. For example, it is noticeable that ChP type B organoids (Figure 2B) are all from experiment 2. Similarly, in the Cer Type B organoids, only 2 out of 9 are from experiment 1. How is this taken into consideration for the data analysis and interpretation of Figure 2? Further experiments using the exact same set-up would be necessary to establish that the difference between type A and type B responses is not due to the sampling strategy.

>We appreciate this comment. We agree that the description of sampling time points is misleading without the information that we did ensure daily exchange of the described volume of media in experiment 2 to ensure the comparability to the first experiment. We added this information to the methods section (Lines 417-419).

>The reviewer's observation on the affiliation of most Type B organoids of both morphologies with experiment 2 is correct. It must be noted, that in our pioneer approach in experiment 1, many organoids were harvested on 2 and 4 dpi, leading to their exclusion for our further analysis, where 7 dpi was set as minimum of sampling to evaluate the course of infection. As experiment 2 had an overall bigger sample size ($n = 27$) of infected organoids in comparison to experiment 1 ($n = 20$), 6 organoids per morphology were evaluated for their course of infection in experiment 1 and almost double the amount, 10 organoids, per morphology in experiment 2. While the inherent heterogeneity of our model does present certain challenges, it enables a broad approach, providing an overview and serving as a foundation for future, more targeted investigations. Nevertheless, this highlights the importance of conducting two individual experiments, when using a model system with expected heterogeneity to gain a more reliable and complete understanding.

5. Another important factor that may affect the replication kinetics is the size of the organoids used for infection. The manuscript reports that organoids were infected with 1×10^4 FFU/organoid. Given the difficulty of establishing MOI for organoids due to their 3D and complex nature, it is important to select organoids of approximately the same size for infection. Have the authors taken this into consideration? Was there a size difference between type A and type B organoids? This will have an impact when comparing Cer with ChP samples, as by the description of the ChP in lines 175-177 of the manuscript, the MOI is much higher in ChP vs Cer organoids. Further direct comparisons between the two models should take this into consideration and should also be mentioned in the discussion.

>We agree that the establishing of MOI for organoids is a complex issue. Therefore, we used organoids of approximately the same size (2–3 mm) for our experiments, expecting no major effects to be caused by difference in organoid size. Additionally, the infection of organoids of the same batch with EBOV virus led to very different replication kinetics with no viral clearance observed (unpublished data Widerspick et al.; manuscript in revision). Hence, while we cannot fully exclude any effect of the organoid size on the course of infection, we conclude that other factors must be the main cause of the differences observed.

[Redacted]

>The MOI can roughly be estimated by cell count via flow cytometry (n = 4 pooled organoids). A cell number of 3.17×10^6 cells was observed. However, it must be noted that this does not account for cell loss due to incomplete or excessive organoid digestion or samples processing. However, using this as a rough estimate the infection with 1×10^4 FFU would account for an MOI of approximately 0.003. The estimated MOI was added to our manuscript (Lines 409-411).

>We see how our description of (original manuscript lines 175-177) can be misleading in this context. It must be clarified, that due to the fluid-filled vacuole characteristic for choroid plexus identities these organoids were of a bigger size overall, however the difference was attributed to the fluid-filled vacuole but not additional tissue. We agree, that the vacuole compartment overall might impact the course of infection as discussed in the manuscript (Lines 334-337). The lines 175-177 of the original manuscript describes a disadvantage in our methodology as the vacuoles, which even though they are comprised of only a small layer of cells can grow bigger than the organoids themselves, collapse during the fixation process performed upon organoid harvest. When processed for sectioning, in which thin ($2 \mu\text{m}$) sections are taken, the collapsed vacuoles introduce a challenge. Additionally, the neurotropism observed is in line with the literature, where neurons are described as the main target, but also astrocytes have been reported to be infected by WNV (Marle et al., 2007 J Virol, PMID: 17670819; Cheeran et al., 2005 J Neurovirol, PMID: 16338745).

Figure 2: Dot plots depicting FSC-SSC gates generated using flow cytometry of dissociated human cerebral organoids of HMGU-1 origin (n=4 organoids pooled).

6. Figure 3 shows staining of WNV in neuron- and astrocyte-rich regions. Is this distribution the same for type A and type B? Please show data for both situations.

> We appreciate the suggestion. However, the course of infection was evaluated in organoids with a minimum of 7 days in the experiment causing organoids of the Type A to be harvested while low titers were detected in the supernatant. The maximum titer of a Type A Cer-organoid detected at harvest was 2.8×10^4 , while the minimum titer at which we detected positive WNV staining was 2.8×10^5 , as described in the manuscript (Lines 103-107). Nevertheless, the staining of neuron- and astrocyte-rich regions is in line with the literature, describing these cell types as WNV targets (Marle et al., 2007 J Virol, PMID: 17670819; Cheeran et al., 2005 J Neurovirol, PMID: 16338745).

7. What is the purpose of Figure 4? It seems misleading to classify acute and post-acute phase when discussing type A and type B together since they have different courses of infection.

> The rationale in Figure 4 of the original manuscript (Figure 3 in revised manuscript) is to gain insights into the innate response of human cerebral organoids to an infection with WNV. Therefore, we include all organoids including various regional identities to gain an overall picture representing the whole brain, instead of choosing a region-specific model. Importantly, as other methods, like the sampling from human patients and also animal models have limitations on the frequency of sampling, we here show a strong advantage of our in vitro model to sample frequently and hence find effects that might otherwise be missed, as we showed for IL-1RA only significantly increased at one time point at 4 dpi (Lines 142-144), and CCL17

increased at 4 and 7 dpi (Lines 131-136). However, we thank the reviewer for drawing attention to the wording and agree that to simply describe the time post infection the classification of acute and post-acute might be misleading, considering the heterogenous courses of infection that are included. We therefore changed this to early and late phase and corrected this figure and the manuscript accordingly.

8. In Figure 5, are the levels normalized to the mock situation of each of the models? This would make it easier to evaluate differences.

>We thank the reviewer for addressing this topic. In our previous manuscript all samples, including mock samples were normalized to a 0-100 scale without normalization to the control samples. Due to control samples of several analytes showing median values of 0, rendering the calculation of a fold change mathematically impossible, or very close to 0, leading to an obscured picture by producing high fold changes simply upon the low control values, this approach was not feasible for our data. However, the reviewers comment inspired us to revise our data visualization to improve the clarity of our data for the reader. The new approach is described in the methods (543-551) and our new Figure 6. The release of biomarkers by different infected groups is now shown as the delta between the median value of those groups to the median values of the respective controls. Thereby, trends are easier to detect, but must still be confirmed with the absolute data showing all time points, as the delta between median values, similarly to a fold change, cannot represent the data heterogeneity often observed in cytokine measurements.

9. Please provide the N of organoids used for analysis for each readout, either in the methods section or in the figure captions.

> We now provide the sample size (as a range from minimum to maximum sample size) for each readout in the figure captions. Furthermore, the exact sample size for each time point of each assay is provided with the raw data.

10. Lastly, the authors claim that the observation of two types of infection course is representative of the ability of the organoids to represent disease heterogeneity. However, given the lack of biological replicates (both at the host and viral strains levels), it is difficult to support this conclusion.

>We do not understand our model as a comprehensive model to represent the disease heterogeneity in its entirety. However, we could show that our heterogenous model has the potential to show two courses of infection, a strong initial viral replication with an increased clearance rate and a relatively long-term infection, making it of interest to study acute infection as well as causes of the long-term sequelae observed in WNV patients and whether they might be attributed to post-inflammatory damage or persistent infection. We agree that future research would benefit from the comparison of various viral strains with varying pathogenicity as well as stem cell lines with the possibility to compare additional factors like the host's biological sex.

Reviewer #3

Major comments

- Line 225, see comment line 162 below. The kinetics shown in figure 1 and used for grouping, do not represent the kinetics observed in figure 2, at least not for B. Instead the “B group” organoids seem to replicate as efficiently as A, initially, but while A organoids show transient infection, the B group organoids remain infected for at least 14 days. Unless the authors have a clear biological explanation for the difference between the 2 “groups”, this distinction seems to be artificial. Therefore the analyses presented starting at line 294, are difficult to interpret.

> We thank the Reviewer for this comment and agree that the schematic grouping graph (Figure 2 in the original manuscript) was not representative for our grouping strategy and changed it accordingly (Figure 4 in the revised manuscript). To provide a clearer understanding of our WNV infection data and rationale for the grouping of our data, we provide a new figure in the revised manuscript (Figure 1), showing not only the heterogeneous viral kinetics, but importantly a bimodal distribution observed in the time point of viral peak. The following grouping depending on this bimodal distribution revealed differences, as an early peak lead to a subsequent decline in titer and was more likely to result in viral clearance, while the late peak was mainly associated with long-term infection. Strikingly, while heterogeneity in the viral titers can also be observed upon the infection with other viruses, such differences in the course of the replication cannot be observed upon infection with Ebola virus (EBOV) or Lassa virus (LASV) of organoids from the same batch (unpublished data Widerspick et al.; manuscript in revision). Here, either persistence or clearance was observed for each virus.

[Redacted]

Minor comment

- Line 78, while the use of 2 different batches controls for some of the biological variability, results are still based on cells from a single donor?

> The results we present in our study are based on the established HMGU-1 human iPSC line. We agree, that future research would benefit from additional experimental components such as cell lines, potentially of different sex etc., or WNV strains of different pathogenicity.

- Line 128, although determining the multiplicity of infection with a complex cells system such as an organoid is difficult, it would be good to indicate whether the dose used, would be high or low, perhaps by providing an average number of cells in organoids.

>Indeed, such a complex system imposes challenges to apply a multiplicity of infection. The MOI can roughly be estimated by cell count via flow cytometry (n = 4 pooled organoids). A cell number of 3.17×10^6 cells was observed. However, it must be noted that this does not account for cell loss due to incomplete or excessive organoid digestion or samples processing as well as a small size variation of the organoids (2-3 mm). To provide the most stable experimental setup possible we decided to focus on the absolute amount of WNV particles. However, using this as a rough estimate the infection with 1×10^4 FFU would account for an MOI of approximately 0.003. This shows a low infectious dose, whereby only about 0.3 % of cells are expected to be infected during the 1 h incubation with WNV. The low infectious dosage is mentioned in the Results section of our manuscript (Lines 75-76). Additionally, we added the estimated MOI to the Methods section (Lines 409-411).

Figure 4: Dot plots depicting FSC-SSC gates generated using flow cytometry of dissociated human cerebral organoids of HMGU-1 origin (n=4 organoids pooled).

- Line 162-164. The rationale for grouping organized based on their replication kinetics is not clear. Does this suggest different stages of development, or is it merely a result of the variability?

> As described above, variability inherit to this model systems might account for some heterogeneity in the course of infection but such bimodal distribution of viral peak time points is not observed in organoids of the same batch when infected with other viruses like EBOV or

LASV (unpublished data Widerspick et al.; manuscript in revision). As we discuss in the manuscript (Lines 326-342), there might be various reasons for these differences. We showed differences in the neuroinflammatory profile when considering the course of infection. Such differences in the response to WNV might contribute to the decrease in viral titer and putative clearance. A trend for an increased release of CXCL10, IL-1RA and sTREM-1 was observed for organoids with an early peak in titer (Type A), independent of the presence of choroid plexus structures. The opposing trend was observed for IL-18 making it of interest for studying long-term WNV infection and potentially WNV persistence. Conclusively, besides the broad explorative approach showing WNV replication in human cerebral organoids, as well as the neurotropism and response by the release of various biomarkers, we acknowledge differences in the course of infection. As the comparison to other viruses suggests this heterogeneity is not accounted for by expected variability. We furthermore observed differences in the neuroinflammatory profile, providing starting points for future WNV research.

- Line 221, see previous comment line 128, why is this considered a low dose, compared to what?

>As previously described, providing an exact MOI is challenging in such a complex model. The MOI can roughly be estimated by cell count via flow cytometry. The cell number of 3.17×10^6 cells ($n = 4$ pooled organoids) would account for an MOI of approximately 0.003 in our experimental setup. Therefore, only about 0.3% of cells are expected to be infected during the 1 h incubation with WNV, showing a low infectious dose. The estimated MOI was added to the Methods section of the revised manuscript (409-411).

- Line 259. Staining for MAP2 and GFAP does not seem to overlap with WNV staining, so why are neurons and astrocytes potential targets? Did you attempt to stain for multiple markers and virus in the same organoid, so MAP2, GFAP AND virus?

> The presented staining was performed in serial sections within the same organoid. Due to technical limitations, we could not stain all markers in one section. As the markers MAP2 and GFAP are found in cell body of the respective cell type and WNV was found to be mainly accumulated perinuclear a full overlap in markers is not to be expected. Nevertheless, improved microscopical analysis of larger parts or even full organoids, would be of interest for future research.

- Line 334, since there are no immune cells involved in WNV infection in this model, either because they are absent, or do not react, can you state that this is inflammation? Or merely the release of proinflammatory cytokines and chemokines, which in vivo would lead to inflammation?

> In line 334 of the original manuscript “we describe the successful establishment of a human cerebral organoid WNV encephalitis model which provided insight into disease pathophysiology in a complex 3D environment including astrocytes and microglia as innate immunocompetent cells”. As this does not include the term inflammation, we are not sure whether a different passage was meant.

>While our model does not include invading immune cells of the periphery, it does include microglia, resident immune cells of the brain, as well as immunocompetent astrocytes. We

show that diverse signaling molecules, including cytokines, chemokines and other markers are secreted into the supernatant as a reaction to the infection with WNV. In our manuscript we discuss the potential origins of these markers as well as their reported involvement in the neuroinflammatory response (Lines 249-303). We acknowledge the inconsistent use of the word inflammation in the literature. This leaves the question of where an exact line would be drawn to separate the response leading to inflammation and the inflammation itself. Based on the literature cited in our manuscript we see our wording justified as we described our cytokines as “pro-inflammatory”. In the lines 230-234 we state, “that sequelae of WNV encephalitis might not only result from late consequences of acute inflammation and thereby inflicted damage, but also from persistent local infection with long-term inflammation. In both scenarios, human cerebral organoids appear well-suited for modeling acute WNV infection as well as WNV persistence in humans.” As studying the response of organoids by the release of pro-inflammatory markers would contribute to studying acute/long-term inflammation, despite this model not accounting for every aspect of these topics, we are satisfied with this wording. However, as stated above we are not sure whether different sections of the manuscript were meant and are happy to engage this discussion again upon clarification, unless of course this already answered the reviewer’s questions.

>Lastly, we do agree, that a further development of our model to include immune cells of the periphery would be of interest in future research as mentioned in the discussion of our manuscript (Lines 348-349).

- Line 433, again the subgrouping and finding of different trends is highlighted, but a discussion on what the biological relevance of these subgroups A and B is lacking.

> As described above, while the heterogeneity might account for some variability in the viral kinetics, the bimodal distribution of viral peak time points is not observed in organoids of the same batch when infected with EBOV or LASV (unpublished data Widerspick et al.; manuscript in revision). As discussed in the manuscript (Lines 326-342), there might be various reasons for these differences. Trends observed in the neuroinflammatory profile upon subgrouping might contribute to the differences observed in those groups, like a decrease in viral titer and putative clearance or long-term infection.

Rebuttal letter

Dear Editor and Reviewers,

We would like to thank you again for the thorough review of our manuscript and rebuttal letter. We thank the reviewers for their positive feedback on the revised manuscript and the additional comments. Below we address the reviewer's remarks and each individual comment point-by-point. The reviewers' comments are reproduced in bold, followed by our responses in plain text. All changes in the revised manuscript are highlighted in blue.

Reviewer #1

The authors have positively addressed most of the concerns raised and improved the quality of the manuscript. In my opinion the main concern at this stage is the biological significance of these results in the context of WNV infection of the CNS. The manuscript mainly reinforces the use of neural organoids to investigate viral infections and highlights variability in replication kinetics and immune response the WNV but does not address underlying mechanisms. Additional points of concern are outlined below:

> In our study, we establish a WNV encephalitis model using human cerebral organoids demonstrating the release of various pro-inflammatory cytokines, chemokines and other biomarkers upon infection, at least similar to what is also seen biologically during human infection (Pavesi et al., *Viruses*. 2024 Feb 29;16(3):3832024 summarizes such findings in humans). However, in this descriptive pilot study, underlying mechanisms were not intended to be the focus of our work, but our results provide interesting starting points and stepping stones for further research.

1- In Figure 3, the authors show and describe in length an inflammatory profile over time upon WNV infection. It is not clear why they chose to do so as it represents all the organoids used in the study which the authors later subclassify as different courses of disease showing distinct profiles. For this reason, I find it misleading to describe a unique profile when this is not the case and suggest that the authors remove this figure from the manuscript.

> We acknowledge the reviewer's point. We nonetheless believe that Figure 3 is of value to the publication, as it shows the *overall* effects observed upon WNV infection of this model. We have confidence that the subsequent approach to subgroup the data does add onto the overall profile. Subgrouping revealed trends that are interesting starting points for future research – for example on regional anatomical identities such as the choroid plexus, or differences in viral replication.

2- It should also be noted, with regards to the inflammatory profile, that the authors did not include a heat inactivated control on these experiments and therefore should discuss the possibility that some of the observed responses, especially acute, may be due to exposure to inactive viral particles and not to active infection. This should be discussed.

> We appreciate the reviewer's comment and have revised the manuscript accordingly (Lines 253-256).

3- In line 153 the authors provide the rational for classification of type A and type B organoids based on the timepoint of peak infectivity. Why is this the only factor considered? In line 176, the authors state that 64% of type A organoids show viral clearing. Why are these not considered a third course of infectivity and constitute a third group? Do these organoids show a distinct immune profile?

> We thank the reviewer for this comment. In addition to the viral peak already we considered the presence or absence of choroid plexus structures as described in lines 163-167 and 320-325. Thereby, our dataset was downsized to a minimum of 6 organoids per group. Considering the heterogeneity of our model, we would like to refrain from further subgrouping of our dataset, resulting in an additional reduction of group sizes. Nevertheless, the reviewer raises an important question for future research and we have added the point to the discussion (Lines 346-349).

4- The authors should address in the discussion the challenges that the model heterogeneity brings to the data interpretation, especially given the low (n=2) number of batches used.

> We thank the reviewer for sharing their thoughts on this and added this point to the manuscript (Lines 349-351).

5- In lines 235-241 of the discussion authors discuss the WNV neurotropism suggesting that neurons and astrocytes are the targets. However, the authors do not demonstrate colocalization of WNV with any of the used cell markers, and therefore fail to show neurotropism, this should be clear in the discussion. Further in lines 241-243, authors suggest that the lack of microglia in the site of infection suggests that they are not directly involved in the response to WNV. Is it possible that the microglia are not functional? This should be addressed.

> In lines 238-241 we describe the cell types present at the site of infection, namely neurons and astrocytes, which both have previously been shown to be susceptible to WNV infection, with neurons being described as the main target. As the cell markers MAP2 (neurons) and GFAP (astrocytes) are found in the cell body of the respective cell type and WNV was found to be mainly accumulated perinuclearly, a full overlap in markers is not to be expected. This point has been added to the manuscript (Lines 113-114). Furthermore, we removed the term neurotropism and now describe the neurons and astrocytes as potential targets in our model (Lines 98, 236). Microglia and astrocytes are known to contribute to inflammatory responses involving an intense immunological cross-talk via cytokines which we have demonstrated to be released in the supernatant of infected organoids. We cannot exclude, however, if microglia, which we did not find directly at the site of infection, are not (fully) functional and added this point to the manuscript (Lines 278-279).